# In Situ Redox Synthesis of Highly Stable Au/Electroactive Polyimide Composite and Its Application on 4-Nitrophenol Reduction

**DOI:** 10.3390/polym15122664

**Published:** 2023-06-13

**Authors:** Yi-Sheng Chen, Wei-Zhong Shi, Kun-Hao Luo, Jui-Ming Yeh, Mei-Hui Tsai

**Affiliations:** 1Department of Chemical and Materials Engineering, National Chin-Yi University of Technology, Taichung 411030, Taiwan; easonchen1224@gmail.com (Y.-S.C.); weizhong9732@gmail.com (W.-Z.S.); 2Department of Chemistry, Chung Yuan Christian University, Chung Li District‚ Tao-Yuan City 32023, Taiwan; g11001302@cycu.edu.tw; 3Graduate Institute of Precision Manufacturing, National Chin-Yi University of Technology, Taichung 411030, Taiwan

**Keywords:** organic-inorganic hybrid, electroactive polyimide, 4-nitrophenol, in-situ reduction

## Abstract

In this study, we developed a series of Au/electroactive polyimide (Au/EPI-5) composite for the reduction of 4-nitrophenol (4-NP) to 4-aminophenol (4-AP) using NaBH_4_ as a reducing agent at room temperature. The electroactive polyimide (EPI-5) synthesis was performed by chemical imidization of its 4,4′-(4.4′-isopropylidene-diphenoxy) bis (phthalic anhydride) (BSAA) and amino-capped aniline pentamer (ACAP). In addition, prepare different concentrations of Au ions through the in-situ redox reaction of EPI-5 to obtain Au nanoparticles (AuNPs) and anchored on the surface of EPI-5 to form series of Au/EPI-5 composite. Using SEM and HR-TEM confirm the particle size (23–113 nm) of the reduced AuNPs increases with the increase of the concentration. Based on CV studies, the redox capability of as-prepared electroactive materials was found to show an increase trend: 1Au/EPI-5 < 3Au/EPI-5 < 5Au/EPI-5. The series of Au/EPI-5 composites showed good stability and catalytic activity for the reaction of 4-NP to 4-AP. Especially, the 5Au/EPI-5 composite shows the highest catalytic activity when applied for the reduction of 4-NP to 4-AP within 17 min. The rate constant and kinetic activity energy were calculated to be 1.1 × 10^−3^ s^−1^ and 38.9 kJ/mol, respectively. Following a reusability test repeated 10 times, the 5Au/EPI-5 composite maintained a conversion rate higher than 95%. Finally, this study elaborates the mechanism of the catalytic reduction of 4-NP to 4-AP.

## 1. Introduction

The rise of industrial civilization has led to the improvement of people’s living standards, but also has led to environmental and human health problems. Among the many water pollutants, 4-nitrophenol (4-NP) is a simple organic aromatic compound that is widely used in dyes, pesticides, and pharmaceuticals [1]. Because of the improper discharge of industries in modern society, 4-NP has become a common pollutant in soil and groundwater. It is worrying that 4-NP is easily dissolved in water and has carcinogenic properties, and many reports have shown that inadvertent ingestion or entry into the human body through the respiratory tract can cause nausea, headache, and eye irritation [2]. In fact, this is one of the reasons why the U.S. Environmental Protection Agency (USEPA) has identified 4-NP and its derivatives as the top priority pollutants for removal [3]. 4-NP is used in various forms in agriculture and other industrial environments, and effective removal and detection of 4-NP is essential due to its adverse effects on humans and the environment.

Several methods of 4-NP removal have been reported in the literature. These methods contain adsorption [4], photocatalytic degradation [5], microbial decomposition [6,7], Fenton [8], peroxymonosulfate [9], and solid phase extraction [10]. Unfortunately, these methods require stringent operating conditions and time-consuming processing, thus increasing the economic demand and possibly causing secondary pollution problems to the environment. So far, catalytic reduction has been considered by many researchers as a relatively green and economical technique for the decomposition and removal of pollutants under mild conditions. The conversion of 4-NP to the less toxic 4-aminophenol (4-AP) by sodium borohydride (NaBH_4_) as a hydrogen source was evaluated as a basic reaction model in aqueous media [11] and easily monitored with high accuracy by UV-Vis spectroscopy. 4-AP is an environmentally friendly and important intermediate for the preparation of many economic products, such as dyes manufacturing [12], photographic developers [13], antipyretic drugs [14], and corrosion inhibitors [15]. However, the catalytic reduction of 4-NP is very difficult and time-consuming in the absence of catalysts due to the need to overcome kinetic barriers [16]. Therefore, the development of highly efficient, environmentally friendly, and reusable catalysts has received close attention from many research groups in recent years.

Noble metal nanoparticles (such as Au, Pt, and Pd) have attracted great interest due to their unique catalytic properties, optical and electronic structures, biocompatibility, and high specific surface area, and they are widely used in biomedicine [17], sensors [18], fuel cells [19], and heterogeneous catalysis [20]. It is worth noting that the remarkable catalytic efficiency of noble metal nanoparticles can be attributed to the highly ordered structure, rapid electron transfer capacity, and large surface area to volume ratio. volume. Although metal/metal oxide nanoparticles exhibit excellent catalytic efficiency, there are limitations in their application to catalysts, such as: (i) their high surface energy and tendency to agglomerate and form larger size nanoparticles, resulting in reduced catalytic performance; and (ii) difficulty in recycling from the reactants to achieve reusability due to their nanometer scale [21]. Tsai et al. [22] prepared the amino-functionalized zirconium phosphate nanosheet decorated with Au nanoparticles composite (Au/ZrP) for the catalytic reduction of 4-NP, and the reaction was completed within 180 s with no significant activity loss through ten consecutive cycles. Das et al. [23] prepared a simple and environmentally friendly method for making a reduced graphene oxide (RGO) nanocomposite decorated with silver nanoparticles (AgNPs) for the reduction of 4-NP to 4-AP within 8 min. These studies showed that loading noble metal nanoparticles onto various carriers (e.g., metal oxides, graphene, polymers, clay, metalorganic frameworks) is an effective way to inhibit their agglomeration and improve the catalytic activity through synergistic effects [24,25].

Since the first introduction of polyacetylene in 1977, conductive polymers (CPs) such as polyaniline, polypyrrole, and polythiophene have received much attention for their environmental stability, low cost, ease of synthesis and special doping/de-doping properties [26]. In recent years, the synergistic effect between polyaniline and noble metal nanoparticles has shown admirable catalytic performance and excellent stability in the field of catalysis [27]. However, polyaniline has poor solubility in common organic solvents, in addition, its electrical conductivity decreases with longer cycle times [28].

To solve these problems, oligoanilines can completely replace polyaniline, which have a similar electronic structure and redox property as polyaniline, and can obtain an ordered molecular structure and good solubility Zheng et al. reported the synthesis and identification of aniline tetramer and aniline pentamer [29]. Qiu et al. developed the synthesis and electrochemical properties of aniline nonamer (Nano-aniline) [30]. In addition, many efforts have been made by researchers to incorporate electroactive aniline oligomers into polymers to form electroactive polymers (EAPs). Wang and Chao et al. reported the development of electroactive polymers containing aniline oligomers on the main and side chains, which were used to study the electrochemical sensing and optical properties of the polymers [31]. Zuo and coworkers synthesized electroactive polyurethane gel containing aniline trimers, which exhibited tunable hydrophilicity, swelling capacity, and biodegradability [32].

On the other hand, electroactive polyimides (EPI) derived from aniline oligomers have excellent mechanical strength, thermal stability, and reversible redox behavior of polyimide. Yeh et al. have published many related studies, such as anticorrosive materials [33], chemical sensors [34], gas separation films [35], and electrochromic [36]. In recent years, electroactive polymers or conductive polymers have been developed and used as promising catalyst carriers. The high specific surface area and microporosity not only confine the growth of metallic nanoparticles, but also highly dispersion compared to conventional catalyst carriers (e.g., graphene, metal oxide) [37]. More interestingly, metal compounds (e.g., HAuCl_4_^−^) have been reported to reduce nanoparticles in situ via electroactive polymers, which provides a convenient way for metal particle catalysts [38]. However, in our previous reports, in situ reduction of metallic nanoparticles by electroactive polymers as catalysts for the reduction of 4-NP has not existed.

In our previous publication, AuNPs were successfully immobilized on polymers with amine functional groups on the electroactive polyamide [39]. Here, we prepared electroactive polyimide based on aniline pentamer (EPI-5) derivatives and configured them with different concentrations of Au ions to obtain Au nanoparticles on the surface of EPI-5 using in-situ redox reactions to form a series of catalysts to evaluate the efficiency and performance for the reduction of 4-NP.

## 2. Materials and Methods

### 2.1. Chemicals and Instrumentation

Aniline (99%, Alfa aesar, Lancashire, UK) distilled before use, ammonium peroxodisulfate (APS) (Thermo Fisher Scientific Inc., Waltham, MA, USA), Pyridine (98.9% Randor, J. T. Baker, PA, USA), N,N-dimethylformamide (DMF, J. T. Baker, PA, USA), N,N-dimethylacetamide (DMAc, Duksan, New Taipei, Taiwan), 4,4′-(4,4′-Isopropylidenediphenoxy)bis(phthalic anhydride) (BSAA, 97%, Sigma-Aldrich, St. Louis, MO, USA), 4,4′-diaminodiphenylamine sulfate hydrate (97%, TCI, Tokyo, Japan), N-Phenyl-p-phenylene diamine (98%, Alfa, Lancashire, UK), sodium chloride (NaCl, Sigma-Aldrich, St. Louis, MO, USA), hydrogen tetrachloroaurate(III)trihydrate (99%, HAuCl_4_.3H_2_O, Thermo Fisher Scientific Inc, Waltham, MA, USA), 4-nitrophenol (99%, Acros, Lancashire, UK), hydrochloric acid (37%, Fluka, NC, USA), acetic anhydride (99%, Fluka, NC, USA), ammonium hydroxide (25%, Fluka, NC, USA), sulfuric acid (97%, SHOWA, Tokyo, Japan), sodium borohydride (99%, Acros, Lancashire, UK), hydrazine (35%, Thermo Fisher Scientific Inc., Waltham, MA, USA).

^1^H-NMR (proton nucleus magnetic resonance spectroscopy, Agilent Technologies DD2, Santa Clara, CA, USA), FTIR spectra (Jasco FT/IR-4600, Tokyo, Japan) were used for the chemical structure of the electroactive material. GPC (gel permeation chromatography, Waters-150 CV, Milford, MA, USA) was used to determine the molecular weight of the sample. Cycle voltammetry (AutoLab, NLD, Utrecht, The Netherlands) was used for electroactive study. UV-Vis spectra (Jasco V-750, Tokyo, Japan) was measure the concentration of 4-Nitrophenol in the reaction during catalytic process and redox performance of the electroactive materials. The morphological were performed using high resolution transmission electron microscope (JEOL JEM-2010, Tokyo, Japan) and scanning electron microscopy (JEOL JSM-7100F, Tokyo, Japan). The formation of reduced AuNPs on the surface of EPI-5 was confirmed by X-ray diffraction analysis (PANalytical X’Pert3 powder diffractometer, Malvern Panalytical, Malvern, UK). The binding energy of the surface phase composition of the electroactive materials was recorded by x-ray photoelectron spectroscope (ULVAC-PHI, PHI 5000 VersaProbe, Chigasaki, Japan). Thermogravimetric analysis (TA Q500, USA) was used to calculate the AuNPs content in series Au/EPI-5.

### 2.2. Synthesis of Amino-Capped Aniline Trimer (ACAT)

Amino-capped Aniline Trimer (ACAT) was prepared by following the procedure reported by Yeh et al. [40]. First, aniline (1.5 g, 16.0 mmol) and 4, -4′-diaminondiphenylamibe sulfate (4.73 g, 16.0 mmol) were dissolved in HCl aqueous solution (1 M, 150 mL) containing 15 g of NaCl. A solution of ammonium persulfate (3.6 g, 16.0 mmol) in HCl aqueous solution (1 M, 25 mL) was added to the previously described solution maintained at 5 °C using a dropping funnel. The mixture was stirred for 1 h at 5 °C. The precipitate was collected through filtration, followed by washing with HCl aqueous solution (1 M, 150 mL). Then, the filtrate was washed with NH_4_OH solution (1 M, 300 mL) for 12 h and followed by washed with large amounts of distilled water. A black powder was further dried in dynamic vacuum oven at an operational temperature of 60 °C for 3 h to obtain ACAT, which was used in the next experiment to synthesize Amino-capped Aniline Pentamer.

### 2.3. Synthesis of Amino-Capped Aniline Pentamer (ACAP)

ACAT(1.5 g, 5.5 mmol) and N-phenyl-p-phenylene diamine(0.96 g, 5.5 mmol) were dissolved in the 50 mL of DMF at 5 °C. A solution of ammonium persulfate (0.91 g, 4.0 mmol) in HCl aqueous solution (1 M, 40 mL) were gradually doped into the above-mentioned solution at 5 °C while maintaining stirring for 3 h, followed by washing with mixture (12 mL DMF/12 mL 1 M HCl/12 mL DI water), precipitated, and collected by filtration, washed by NH_4_OH solution (1 M, 500 mL) for 12 h and followed by washed with large amounts of distilled water. A black powder was further dried in a vacuum oven at 60 °C for 5 h to obtain Amino-capped Aniline Pentamer (ACAP).

### 2.4. Synthesis of Electroactive Polyimide (EPI-5)

The electroactive polyimide (EPI-5) was synthesized using ACAP and BSAA. First, BSAA (0.52 g, 1.0 mmol) was added to 8.0 g of DMAc at room temperature with continuous stirring for 30 min. A solution containing ACAP (0.47 g, 1.0 mmol) in another 8.0 g of DMAc were prepared under magnetic stirring at room temperature. Dianhydride of BSAA was reacted with ACAP, followed by stirring for 24 h to generate electroactive poly (amic acid) (EPAA-5). The EPAA-5 was converted to electroactive polyimide by chemical imidization reaction. The reaction was done by adding the mixture of acetic anhydride/pyridine (0.102/0.079, *v*/*v*) to the previous solution containing EPAA-5 under stirring for 1 h and refluxing for 3 h under nitrogen. The as-prepared EPI-5 solution was slowly added dropwise to an excess amount of methanol to precipitate the product. After drying at 60 °C in the vacuum for 3 h to obtain electroactive polyimide based on ACAP (EPI-5).

### 2.5. Preparation of Series of Au/Electroactive Polyimide (Au/EPI-5) Composite

The series of Au/electroactive polyimide (Au/EPI-5) were prepared by the in-situ redox reaction between the EPI-5 and HAuCl_4_. First, 0.1 g as-prepared EPI-5 powder was dispersed into 10 mL of 1M NH_4_OH containing 3 mL of hydrazine solution. The mixture was stirred for 24 h, filtered, and washed with water until pH became neutralized, followed by freeze-drying treatment at −42 °C for 24 h. EPI-5(leucoemeraldine base) was collected in form of black powder. Series of Au/electroactive polyimide (Au/EPI-5) was prepared by immersing 0.1 g EPI-5(lucoemeraldine base) in 25 mL of different concentration (1, 3, 5 mM) HAuCl_4_•3H_2_O for 6 h. The Au/EPI-5 powder was subsequently collected by centrifugal filtration and followed by washing with excess amount of distilled water and freeze-drying treatment at −42 °C for 24 h, the as-prepared series Au/EPI-5 composites was collected as black powder. The synthetic route for EPI-5 decorated with Au nanoparticles is shown in Figure 1.

### 2.6. Electrochemical Cyclic Voltammetry of EPI-5 and Series of Au/EPI-5 Composite

In this study, the redox property of as-prepared EPI-5 and series of Au/EPI-5 composites. EPI-5 and series of Au/EPI-5 composites were determined by coating the materials onto glass carbon electrode (working electrode), followed by a series of electrochemical cyclic voltammetry (CV) study. 0.01 g of EPI-5 and series of Au/EPI-5 composites were dissolved in 3 mL DMAc. These solutions were coating on the glass carbon electrode and dried at room temperature.

### 2.7. Catalytic Activity

To study the EPI-5 and series Au/EPI-5 composites were applied in the reduction reaction of 4-NP. NaBH_4_ was used as a hydrogen source in water. 0.5 mg EPI-5 and series Au/EPI-5 composites were dispersed in 3 mL of 4-NP. Then prepared 0.3 mL NaBH_4_ solution (100 mM) was introduced to the above 4-NP solution. And the time-dependent absorbance was recorded by UV-Vis absorption spectra. Finally, the kinetic rate constant, activation energy, and reusability of 4-NP were evaluated.

## 3. Results

### 3.1. Characterization of ACAT, ACAP, EPAA-5, and EPI-5

The representative ^1^H NMR spectra of ACAT and ACAP are shown in Figure 1 (includes subfigures of ACAT and ACAP chemical structures) to confirm the chemical structure. From Figure 1a, the spectra of ACAT reveals the signal at 5.42 ppm and 7.03 to 6.48 ppm corresponded to the primary amine protons (-NH_2_) and aromatic protons [40]. The signals of ACAP shows the signal at 5.41 ppm shown in Figure 1b could be also assigned to the primary amine protons(-NH_2_), and the signals around 7.03 to 6.60 ppm represents the splitting of aromatic protons [41].

In order to further confirm that ACAT and ACAP have been successfully synthesize, as-prepared ACAT and ACAP were characterized by FTIR, as shown in Figure 2. The results show the ACAT and ACAP have the same characteristic peak at 3309 and 3205 cm^−1^ corresponding to the -NH_2_ group. Moreover, the characteristic peak at position of 1598, 1498 cm^−1^, and 1596, 1500 cm^−1^ may be due to the vibration bands of quinoid rings and benzenoid rings of ACAT and ACAP, respectively. The characteristic peak of C-N on the amine group was detected at 1274 cm^−1^. Finally, the characteristic peaks were found at 834, 736, 697 cm^−1^ [41], which attributed to the bending of C-H group of the hydrogen atom at the benzene. These characteristic peaks of overall spectra indicated the ACAT and ACAP were successfully achieved.

As shown in Figure 2c,d, FTIR spectra used to characterize the structure of EPAA-5 and EPI-5 to further confirm the successful preparation. The FTIR spectra of EPAA-5 shows the main characteristic peak at 3200–3400 cm^−1^, corresponding hydroxy group (O-H) and amine group (N-H), and the carboxylic acid group (C=O) appeared at 1707 cm^−1^. In addition. The characteristic peak at 1598 and 1498 cm^−1^ were assigned to vibration bands of quinoid rings and benzenoid rings of EPAA-5, respectively. After reacting with anhydride/pyridine solution, the EPAA-5 was converted into EPI-5 by performing the chemical imidization. the characteristic peak of EPAA-5 at the position of 3200–3400 cm^−1^ was found to completely disappear, as shown in Figure 3. On the other hand, EPI-5 was found the asymmetric and symmetric carbonyl stretching vibration peak at 1776 and 1714 cm^−1^ [42]. EPI-5 was also observed at 1595 and 1500 cm^−1^, which represented the characteristic peak of quinoid rings and benzenoid rings. Based on the characteristic peaks of FTIR spectra, it indicated the complete conversion of EPAA-5 to EPI-5 by the chemical imidization reaction.

### 3.2. Chemical Oxidation of EPI-5

In this experiment, UV-Vis spectra can determine the redox properties of the EPI-5. The as-prepared leucoemeraldine base (fully reduction state) of the EPI-5 was dispersed in a DMAc solution. Subsequently, trace amounts of oxidant agent, (NH_4_)_2_S_2_O_8_, were gradually added to the EPI-5 solution reach the pernigraniline base (fully oxidation state), which was continuously monitored every 180 s of the sequential oxidation process of the EPI-5 in UV-Vis spectra, as shown in Figure 3. Firstly, two absorption peaks were appeared at position at 315 and 575 nm, which were associated with the π–π* transition of the conjugated ring system and the transition between the benzenoid ring and quinoid ring [43], respectively. After the addition of trace amounts of the oxidation agent, slow oxidation of ACAP segments in EPI-5 was observed. The absorption peaks found in UV-Vis spectra started a blue shift from 315 nm to 310 nm, which results accompanied by a decrease in intensity. At the same time, a new absorption peak appeared at 575 nm to 570 nm [44], which was related to the excitation-type transition between the HOMO orbital of the benzenoid ring and the LUMO orbital of the quinoid ring. Possible mechanism for this behavior can be interpreted as follows: Initially, EPI-5 reaches the LEB state of the ACAP segment through the reducing agent, hydrazine, and there is no quinone ring in the state. After adding a small amounts of oxidation agent, it reached first EB state with each ACAP segment containing only one quinone ring in this state. Finally, EPI-5 was oxidized to a PNB state with each ACAP segment containing two quinone ring in this state, it exhibited a blue shift, as shown in the inset in Figure 4. These results confirmed the chemical oxidation process of EPI-5.

### 3.3. Chemical Structural and Morphological Characterization of EPI-5 and Series Au/EPI-5

#### 3.3.1. Characterization of EPI-5 and Series of Au/EPI-5 Composites by FT-IR

Figure 4 show the representative FT-IR of the EPI-5 and series Au/EPI-5 composites, respectively. FT-IR spectra of EPI-5 and series Au/EPI-5 showed the characteristic peaks around 1776 and 1714 cm^−1^ corresponding to asymmetric and symmetric carbonyl stretching vibration. The characteristic peaks around 1595 and 1500 cm^−1^ are attributed to the stretching vibration of C=C in the quinoid (Q) and benzenoid ring (B), respectively. These results indicate that the in-situ redox reaction between Au ions and EPI-5 is reduced on the surface of EPI-5 without changing the structure.

It should be noted that, with the increase of reduced AuNPs in the composites, the intensity of C=C stretching vibration for quinoid rings increases obviously. in addition, the intensity of C=C stretching vibration for benzenoid ring decrease was observed. Ascribed to the redox reaction between Au ions and EPI-5, leading to a decrease in the intensity ratio of benzene rings (B) to quinone rings (Q) with increasing amounts of reduced AuNPs [44]. Indicating the ratio of benzene rings(B) to quinone rings (Q) according to Table 1.

The TGA analysis curves of EPI-5 and series of Au/EPI-5 composites in an air atmosphere and shown in Figure 5. The main purpose is to completely creak the electroactive materials and determine the loading of Au on the EPI-5 surface. After the process, the residue contents of EPI-5 and series of Au/EPI-5 were 1.44%, 6.17%, 12.30% and 18.90%, respectively, which can be utilized to calculate the Au content in the Au/EPI-5 composite. Indicating the loading of Au according to Table 1.

The XRD patterns of the EPI-5 and series of Au/EPI-5 composites are shown in Figure 6. It can be seen that as compared to EPI-5 and series of Au/EPI-5 composites showed four additional characteristic peaks at 2θ of 38.1°, 44.3°, 64.5°, and 77.4°,which are ascribed to the (111), (200), (220), and (311) crystallographic planes of Au [45], respectively. This result shows that the AuNPs were reduced and anchored on the EPI-5 surface, the intensity of the characteristic peaks also increases as the amounts of reduced AuNPs increase.

The surface chemical composition of the EPI-5 and series of Au/EPI-5 composites was also investigated by XPS analysis. Figure 7 displays the full survey scan spectra and high solution XPS spectrum of various elements present in all materials. The survey scans spectra of EPI-5 and series of Au/EPI-5 reveal clear signals of Au 4f, C 1s, N 1s, and O 1s at 83, 283, 398 and 530 eV, respectively. The spectra in Figure 7b display two diffused peaks with binding energies of 82.2–85.5 and 85.9–86.1 eV, which correspond well with Au 4f_7/2_ and Au 4f_5/2_ spin-orbit splitting of metallic gold (Au^0^) [46], respectively. The appearance of Au0 is attributed to the redox reaction between oligoaniline segments and Au ions. Moreover, some reports that the interaction between mental nanoparticles and conductive polymer composite, Yang and coworkers [47] studies the Fe-PANI composite catalyst and explore the interaction between the Fe atom and N groups to reveal the catalytic site more clearly on Fe-PANI. Getting or losing electrons would certainly influence the chemical structure of electroactive polymers due to their redox properties. Therefore, with the electron transfer between AuNPs and EPI-5, the oxidation or the reducing state of EPI-5 would be changed. As shown in Figure 7c,d, the N 1s core-level spectra of EPI-5 and 5Au/EPI-5 were determined to further reveal the different chemical states of the N groups. The three different electronic states peak at 397.4–397.1, 398.9 and 402.0–401.4 eV were assigned to the benzenoid amine (-NH-), quinoid amine (=NH-), and nitrogen cationic radical (N^+^) groups [48,49], respectively. Quantification and identification of N species over EPI-5 and 5Au/EPI-5 for N 1s core-level spectra data were summarized in Table 2. The ratio of N^+^ species (sum of =NH- and N^+^) to N species (-NH-) was calculated to evaluate the electron transfer between AuNPs and EPI-5. The N^+^/N ratio was 0.56 and 2.74 for the EPI-5 and 5Au/EPI-5, respectively. It means that after reducing the AuNPs on the EPI-5 surface, EPI-5 loses a lot of electrons and was oxidized by in situ redox reaction between Au ions and EPI-5. Similar to the FTIR results, the content of =N- was increased after the loading of Au, which indicated that AuNPs coordinated with the -N= group are more energy favorable than that with the -NH- group. According to the XPS analysis results, the possible series of Au/EPI-5 composite formation process is consistent with the XRD and FTIR results.

The morphologies of the EPI-5 and series of Au/EPI-5 composites were characterized by SEM and HR-TEM, as shown in Figure 8. Compared with the morphology of the EPI-5, the AuNPs were clearly observed in the series of Au/EPI-5 composites, which showed that they were regularly dispersed. After loading the AuNPs, many black dots with average particle size in the range of 23–113 nm (as shown in the inset image of Figure 8f–h) can be observed on the EPI-5 surface, which indicates that AuNPs were anchored on the EPI-5, the particle size of the reduced AuNPs increases with the increase of the concentration.

#### 3.3.2. Electroactive Properties of EPI-5 and Series of Au/EPI-5 Composites by Electrochemical CV Studies

In this study, all as-prepared EPI-5 and series of Au/EPI-5 composites were measured by cyclic voltammograms (CV) using the three-electrode electrochemical cell in 40 mL H_2_SO_4_ solution at a scan rate of 50 mV·s^−1^, as shown in Figure 9. The results show the bare GCE did not exhibit a redox peak. The CV curve of EPI-5 was observed with a relatively small peak of oxidation current at 0.78 μA. Moreover, the AuNPs were reduced and anchored on the EPI-5 surface showed in an increasing in redox current. The series of Au/EPI-5 showed an increase trend: 1Au/EPI-5 < 3Au/EPI-5 < 5Au/EPI-5, this current enhancement phenomenon indicates the AuNPs transforming higher electron transfer [50]. As we expected that the reduction of the higher concentration of AuNPs on the surface of EPI-5 do effectively enhance the redox capability, which is in line with FTIR, TGA, XRD, XPS, and HR-TEM analysis.

### 3.4. Catalytic Characterization of EPI-5 and Series of Au/EPI-5 Composites

To evaluate the catalytic activity of the prepared EPI-5 and series of Au/EPI-5 composites, the reduction of 4-NP to 4-AP has been chosen as a model reaction in the presence of excess NaBH_4_, which can be monitored by time-dependent UV-Vis spectroscopy. The absorption peak at 317 nm was attributed to the 4-NP solution. During the experiment, after addition of freshly prepared NaBH_4_ in the 4-NP solution, the phenolic hydroxyl group in 4-NP loses a proton, 4-NP ions are formed, and the absorption peak was shifted to 400 nm. 10 min after addition of the EPI-5 to the mixed solution of NaBH_4_ and 4-NP, the absorption peak was hardly change at 400 nm [51]. Notably, EPI-5 had the lower catalytic activity of catalytic reduction of 4-NP to 4-AP, which acted as catalyst carriers, as shown in Figure 10.

However, when adding a series of Au/EPI-5 composites (as shown in Figure 11), a new absorption peak was observed at 300 indicating the formation of 4-AP. At the same time, the absorption peak reduces of 4-NP with increasing peak strength of 4-AP. The completely reduction of 4-NP to 4-AP take about 17 min when 5Au/EPI-5 is used as the catalyst, it shows the best catalytic activity. And for 1Au/EPI-5 and 3Au/EPI-5, which are take about 43 min and 25 min. these results show the AuNPs is an essential substance for catalyst, and their characteristics affect the catalytic activity [52,53]. We can get the order of catalytic activity: 5Au/EPI-5 > 3Au/EPI-5 > 1Au/EPI-5, which may be attributed to the successful loading of Au (reference TGA analysis) and the electronic metal-support interactions between the EPI-5 and AuNPs. Specifically, the absorbed BH_4_^−^ donates electrons to the AuNPs, and the electron transfer between EPI-5 and AuNPs would facilitate the electron donation of BH_4_^−^. Additionally, Au^0^ was confirmed in the EPI-5 by XPS analysis, and they could act as electron acceptor facilitating the electron transfer and the reduction of 4-NP, which endows the catalyst with excellent catalytic stability [54].

As the concentration of 4-NP was much lower than the NaBH_4_, the reduction of 4-NP to 4-AP can be regarded as the pseudo-first-order reaction kinetics. The rate constant was obtained using the following Equation (1) [55]:(1)$$ln\frac{C_{t}}{C_{0}}=ln\frac{A_{t}}{A_{0}}=-k\cdot t$$
where *C_t_* and *C*_0_ are the concentration of 4-NP at time *t* = 0 and *t* = *t*, respectively, which are corresponding to absorptance (*A_t_* and *A*_0_) at 400 nm, and *k* is the rate constant. As shown in Figure 12a, the rate of the reaction were determined from the slopes of the linear relation plot of *ln* (*C_t_*/*C*_0_) versus t using the series Au/EPI-5 composites. The values of *k* for 1Au/EPI-5, 3Au/EPI-5 and 5Au/EPI-5 are 7.0 × 10^−4^ s^−1^, 9.0 × 10^−4^ s^−1^, and 1.1 × 10^−3^ s^−1^, respectively. It was evident the 5Au/EPI-5 showed higher reactivity than the other Au/EPI-5 composites, the rate constant values were found to increase with increase loading of Au. In addition, based on the Arrhenius equation [56], the activation energy (*E_a_*) for the reaction was determined using the following Equation (2):(2)$$\ln k=\ln A-\frac{E_{a}}{R}\left(\frac{1}{T}\right)$$
where *k* is the rate constant at different temperature T(K), *A* is the Arrhenius factor, and *E_a_* is the activation energy. As shown in Figure 12b, *E_a_* can be obtained from the slope of *lnk* versus 1/*T*. The *E_a_* of the 5Au/EPI-5 was determined from the Arrhenius plot to be 38.9 kJ/mol.

In addition to the careful catalytic studies, reusability of heterogeneous catalysts is important properties and vital factor of treating pollutants. In a recycle experiment, the 5Au/EPI-5 composite was carefully collected and used for multiple catalytic reactions after waiting for the catalyst to settle slowly for one day after the reduction reaction. As shown in Figure 13, after ten cycles, the conversion of 4-NP still above 95% which indicates the 5Au/EPI-5 developed in the current study shows excellent catalytic stability and robust even a after multiple uses.

The rate constant and the recyclable of the 5Au/EPI-5 composite comparable to those of most catalyst using conjugate polymer as a carrier in the literature and previously published reports (Table 3). The catalytic activity and reusability observed in this work were better than those observed in several reports.

### 3.5. Possible Reduction Reaction Mechanism for 4-NP

To explain the process of the catalytic reduction of 4-NP to 4-AP using the series Au/EPI-5 composites, a possible mechanism for the reduction of 4-NP was proposed based on the Langmuir-Hinshelwood mechanism (as shown in Figure 2). The ionization of NaBH_4_ in the liquid results in the production of BH_4_^−^ and their adsorption AuNPs on the EPI-5 to form hydride complex (Au-H species). At the same time, 4-NP ions adsorb on the hydride complex surface. Then Au-H species are transferred to the adsorbed 4-NP and lead to the reduction of -NO_2_. After three step catalytic hydro-dehydration reaction to generate the corresponding 4-AP. Finally, 4-AP desorbs form the catalyst [60,61].

## 4. Conclusions

In this work, an electroactive polyimide decorated with AuNPs (Au/EPI-5) without the addition of any reducing agent was successfully synthesized and reduced 4-NP to 4-AP. The AuNPs were loaded on the Au/EPI-5 by the in-situ redox reaction between the EPI-5 and HAuCl_4_. Through the comparison of catalytic performance of series Au/EPI-5, we find the 5Au/EPI-5 exhibited the best catalytic activity, which could be attributed to the high loading of AuNPs and the synergistic effect between the EPI-5 and AuNPs. Results showed the rate constant, activation energy of the 5Au/EPI-5 catalyzed reduction of 4-NP were 1.1 × 10^−3^ s^−1^ and 38.9 kJ/mol, respectively, and after 10 cycles, the 5Au/EPI-5 composite maintained at 95% above. In summary, this Au/EPI-5 composite demonstrates many advantages, such as reduction using very low catalyst amount (0.5 mg), remarkable catalytic performance, and excellent recyclability. Making electroactive polymers as a carrier for AuNPs attractive for the catalytic field.

## Data Availability

The data presented in this study are available on request from the corresponding author.

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
