# Peer review of "In Situ Redox Synthesis of Highly Stable Au/Electroactive Polyimide Composite and Its Application on 4-Nitrophenol Reduction"

_polymers, 2023, doi:10.3390/polym15122664_

Round 1

Reviewer 1 Report

Author must rewrite the MS to make it more appropriate and concise. Unnecessary too many citation, which could have been easily avoided. Number of figures are quite high for a single article. The MS looks like a project report for a Master student, rather than a scientific journal. Author must rewrite the MS with more focus objective and intent. 

English need to be improved. 

Reviewer 2 Report

Article Polymer-2423061 demonstrated the synthesis of aniline pentamer based electroactive polyimide (EPI-5) and decorated its surface with different concentration of Au nanoparticle. The Au electroactive/EPI-5 nanocomposite was then utilized for in-situ reduction of 4-nitrophenol to 4-aminophenol. 4-nitrophenol is a common pollutant and exhibit carcinogenic properties, which enters to the environment from agriculture and other industries. It can be easily dissolved in the water and could end up in the human body.  Therefore, its removal from the environment through its conversion to a less toxic product 4-aminophenol is efficient and significant. Therefore, this article can be considering publishing in Polymers after the revision noted below.

1.      ACAT and ACAP are synthesized and characterized. However, only ACAP was utilized for the synthesis of EPI-5 which was used as a catalyst for the conversion of 4-NP to 4-AP. ACAT was not further applied. Therefore, from the article it is not clear that why ACAT was synthesized and characterize?

2.      ACAT is a tetramer, but Figure 1a deficit its structure as trimer.

3.      The language of section 3.1 FTIR discussion is monotonous.

4.      Check and authenticate the quality of Figure 11.

5.      Include the absorption of spectrum of 4-AP at different concentration, so one can understand how much 4-AP are formed. And thus, the true conversion can be calculated. As it seems that the peak at about 300 nm is belong to 4-AP. So, as the reactant 4-NP peak intensity reduces the peak intensity of the product shall increase as discussed in the earlier reports 10.1039/c8ra04332a,  10.1016/j.jenvman.2022.115292.

minor editing of English language is required. 

Reviewer 3 Report

Manuscript ID: Polymers-2423061

I have read the manuscript entitled “In situ redox synthesis of high stable Au/electroactive polyimide composite and its application on 4-nitrophenol reduction”. In this article the authors have described the synthesis of a series of Au/electroactive polyimide (Au/EPI-5) composite for the reduction of 4-nitrophenol (4-NP) to 4-aminophenol using NaBH4 as a reducing agent at room temperature (4-AP). Overall, a larger audience might find the work interesting. To enhance the scientific quality, the writers should address the issues listed below. This work may be taken into consideration for publishing once the indicated improvements have been thoroughly addressed.

1.      Replace “high stable” with highly stable in title of the manuscript.

2.      To attract readers, the abstract should be more specific about scope, goals, and conclusions.

3.      Au is a costly metal. If it was possible to use a low-cost metal instead of Au? Why authors ignored this?

4.      The synthesis process of ACAT and ACAP are quite complicated and involve many chemicals. What about the toxicity effects of the chemicals used and cost?

5.      Label the FTIR spectra (in Fig. 3) and add proper supporting references for explanation.

6.      “The series of Au/EPI-5 showed an increase trend: 1Au/EPI-5 < 3Au/EPI-5 < 5Au/EPI-5, this current enhancement phenomenon indicates the AuNPs transforming higher electron transfer”. Then it is possible that higher Au concentrations may have higher electron transfer and higher photocatalytic activity? Why the optimum Au concentration was not found?

7.      In recycle experiment of the 5Au/EPI-5 composite, the reported results (Fig.15) don’t have a regular trend. So, repeat the experiment.   

8.      Rephrase the conclusion, make its language more proper.

9.      Provide a comparison with previous studies reported in literature.

10.  Some errors regarding the capital/small alphabet, sub/super script, spacing and typo need to consider throughout the manuscript. 

Minor Editing

Reviewer 4 Report

Comments to the Authors:

In this manuscript authors synthesized Au/electroactive polyimide (Au/EPI) composite for the reduction of 4-nitrophenol to 4-aminophenol using NaBH4 as a reducing agent at room temperature. This research has value for the researchers in the related areas. However, the paper needs minor improvement before acceptance for publication. My detailed comments are as follow:

1.      The abbreviation should be placed after name of the compound like 4-aminophenol (4-AP)

2.      In the introduction section, authors should introduce following relevant articles related to catalytic reduction of 4-NP.

a.      doi.org/10.1007/s11164-020-04165-0

b.      doi.org/10.1016/j.nanoso.2023.100960 and others also.

3.      Why authors not selected AgNPs and others low cost materials in comparison to that of AuNPs?

4.      Error bar is missing in Figure 15.

5.      There are typos and grammatical errors.

6.      Why there is increase in catalytic activity after 5th cycle?

7.      Authors should marked the characteristic peaks in the Figure 3

8.      Compare its catalytic activity with other commercially used catalyst.  

Minor editing of English language required
